# Impact of COVID-19 on unmet needs for healthcare in Peru: an interrupted time series analysis

Rodrigo Vargas-Fernández[1]*, Akram Hernández-Vásquez[1], Shanquan Chen[2], Francisco Diez-Canseco[1], Pamela Smith[3], Xanthe Hunt[4,5], Lena Morgon Banks[2]

**1** CRONICAS Center of Excellence in Chronic Diseases, Universidad Peruana Cayetano Heredia, Lima, Peru, **2** Department of Population Health, International Centre for Evidence in Disability, London School of Hygiene & Tropical Medicine, London, United Kingdom, **3** Sociedad y Discapacidad SODIS, Lima, Peru, **4** Africa Health Research Institute (AHRI), Somkhele, South Africa, **5** Institute for Life Course Health Research, Department of Global Health, Stellenbosch University, Cape Town, South Africa

* jrodrigov1803@gmail.com

## Abstract

The COVID-19 pandemic affected people's health and access to healthcare world-wide. Still, little is known about how distinct phases of the pandemic – namely the lockdown period and the period directly after lockdown – impacted these outcomes, and whether there were differences in these impacts among different segments of the Peruvian population. The aim of this study was to examine the impact of different phases of the COVID-19 response on unmet needs for healthcare across Peru. Using an interrupted time series design, this study analyzed quarterly data from Peru's National Household Survey on Living Conditions and Poverty (ENAHO) from 2018 to 2022. Outcome variables included people reporting having a health problem in the last 4 weeks and, among those reporting a problem, whether they had needed and sought care. These outcomes were stratified by demographic characteristics (i.e., by age group, gender, disability, indigenous status, regions, and rural/urban). Step and trend changes in outcomes during and directly after lockdown periods were compared to the pre-pandemic period. Overall, the prevalence of health problems decreased by 18.4 percentage points (pp) during the lockdown period compared to pre-pandemic levels, although unmet healthcare needs increased by 16.7 pp. Once lockdowns were removed, prevalence of health problems and access to healthcare increased slightly, but remained below pre-pandemic levels. Certain groups, namely people with disabilities, indigenous people and people in rural areas, had persistently worse health and poorer access to care. Persistent disparities in unmet healthcare needs highlight the necessity of targeted interventions to address systemic barriers faced by vulnerable populations and ensure continuity of care during health crises.

**Data availability statement:** The ENAHO databases are publicly accessible and can be downloaded from the National Institute of Statistics and Informatics website (https://proyectos.inei.gob.pe/microdatos/). The ENAHO databases can be accessed by visiting the survey query tab, selecting the "ENAHO Metodología ACTUALIZADA", "Condiciones de Vida y Pobreza - ENAHO," and specifying the year of the survey.

**Funding:** The research was funded by the UK Foreign, Commonwealth and Development Office under 2 grants (PENDA project - PO8073 to LMB; COVID Collective - 102866EH to LMB). Part of LMB's time and Open Access is covered by the Arts & Humanities Research Council (AH/X009580/1 to LMB). The funders had no role in study design, data collection and analysis, decision to publish, or preparation of the manuscript.

**Competing interests:** The authors have declared that no competing interests exist.

## Introduction

Sustainable Development Goal (SDG) 3 seeks to ensure "good health and well-being" for all [1]. A critical part of the SDGs and the broader global health agenda is the fulfilment of Universal Health Coverage (UHC), whereby all individuals have timely and affordable access to required health services. However, unmet healthcare needs persist, due to barriers such as high costs linked to seeking and receiving care, and poor availability and quality of required services [2].

The COVID-19 pandemic increased barriers to seeking and providing healthcare globally. Health systems shifted their prioritization of human and other resources to COVID-19 containment and treatment [3,4]. In many settings, particularly in low- and middle-income countries, health systems were already under-resourced pre-pandemic, and so COVID-19 caused additional strain on service provision [5]. Further, movement restrictions, fear of infection and decreased capacity to pay could have affected health-seeking behaviour [6,7]. As such, there were increased barriers to both providing and accessing many routine healthcare services, including primary care and diagnostic, preventative, rehabilitation, and treatment services [8]. A systematic review conducted in August 2020 found that health utilisation decreased by over a third globally during the initial stages of the pandemic [4]. However, only 4 of 81 included studies were from low- or middle-income countries. Additionally, little evidence is available on whether utilisation declines were due to decreased need (e.g., decrease in other infectious diseases, accidents as a side effect of restrictions) or increased barriers to access, and how unmet health needs differed across the population (e.g., by gender, rural vs urban, age groups).

Peru was severely affected by COVID-19, with the highest mortality rate globally at certain time points early in the pandemic [9–11]. Even before COVID-19, about half the population experiencing a health problem reported unmet access to needed healthcare [12]. The lockdown in Peru was one of the earliest and strictest, causing many primary care facilities to reorient towards COVID-19 care. This shift resulted in severe shortages of hospital beds in both general wards and intensive care units [11]. Further, seven million Peruvians became unemployed in 2020, increasing financial barriers to seeking care [13]. Since the end of lockdowns, the healthcare system has struggled to recover, due, for example, to a shortage of healthcare professionals, fragmented financing, a scarcity of medical resources necessary for patient care or the inadequate government response to address the pandemic's challenges [14,15].

Barriers to accessing needed healthcare may be particularly high for certain groups, such as people with disabilities, older adults, people living in rural areas and indigenous groups [16,17]. For example, people with disabilities often have higher unmet health needs compared to people without disabilities as many of the services they require are urban-based and high cost (e.g., rehabilitation, specialist services), health services are often not accessible, or they face discrimination from healthcare providers [18]. Moreover, people living in rural areas of Peru, especially in the Jungle regions, face significant geographic and cost barriers to accessing health care due to disparities in the availability and quality of services [19]. In addition, the

indigenous population has historically faced limited access to healthcare and had particularly high COVID-19 morbidity and mortality [14].

The aim of this study was to examine the impact of COVID-19 on unmet needs for healthcare across Peru. It uses an interrupted time series design to assess how unmet healthcare needs were affected by the imposition of lockdowns in the initial phase of the pandemic. It also explores whether prevalence of unmet health needs have returned to pre-pandemic levels once lockdown measures were removed. Finally, this study evaluates whether impacts of COVID-19 on unmet health needs differ across groups, such as by gender, indigenous status, disability, age, area of residence (i.e., urban, rural), and region.

## Materials and methods

### Data source

An interrupted time series study was conducted using quarterly data from the National Household Survey on Living Conditions and Poverty (ENAHO) during the period from 2018 to the second quarter of 2022. ENAHO is a population-based survey developed and conducted by the National Institute of Statistics and Informatics (INEI) of Peru. It is national representative, including urban/rural areas, the 24 departments, and the Constitutional Province of Callao. This survey is used in Peru to obtain the official data for measuring poverty in the country and collects information on household characteristics, sociodemographic information of household members, such as education, health and healthcare use, employment, and enrolment in social programmes [20].

### Sampling and data collection

ENAHO has a probabilistic, stratified, multistage, and independent sampling design in each study department, which consists of three sampling units according to the area of residence. In urban areas, the primary, secondary, and tertiary sampling units are urban population centers with 1,200 or more inhabitants, housing clusters, and individual household, respectively. In rural areas, the primary, secondary, and tertiary sampling units are the urban population centres with 500–1,200 inhabitants, the rural registration area, housing clusters, and individual households, respectively [20].

The data collection method for ENAHO varied according to the survey year, with some changes due to COVID-19. In 2018 and 2019, data collection was conducted through face-to-face interviews by trained personnel. In 2020, during the period of lockdown, data collection was done through mixed methods: telephone interviews (from March 16 to September 30, 2020) and later, face-to-face interviews complemented with telephone interviews (from October to December 2020). In 2021 and 2022, INEI conducted data collection similarly to the year 2020, using mixed interviews (telephone and face-to-face). Further methodological details regarding the design, instruments, procedures, and manuals for different years of ENAHO can be found in the "Documentación" tab on the INEI web platform [21].

### Inclusion and exclusion criteria

The present study included participants of all age groups who had complete data on the variables of interest. A total sample of 483,093 was included (110,650; 110,294; 107,326; 104,293 and 50,530 participants for the years 2018, 2019, 2020, 2021 and 2022, respectively). The average response rate in each year of the ENAHO was 94·3%, 94·2%, 94·0%, 94·2%, and 94·3% for 2018, 2019, 2020, 2021 and 2022, respectively [22].

### Variables

**Outcome variables.** The main outcome variables of interest were:

i. "Presence of a health problem" included people who reported experiencing any symptoms, discomfort, illness, relapse of chronic illness, or accident in the last 4 weeks.

ii. "Unmet need for healthcare": Among people with a health problem, people were classified as having required but not sought healthcare if they either a) did not seek healthcare and their reason for not seeking it was any reason other than reporting he/she did not need it; or b) sought healthcare but at a non-biomedical facility (i.e., went somewhere other than a healthcare facility of the Ministry of Health, Social Security of Peru (EsSalud), Armed Forces and National Police, private medical service, or pharmacy). For example, people who self-treated at home or sought non-medical care (e.g., traditional healer) were considered as having not sought care.

**Stratification variables.** Differences by individual characteristics were also explored to see if presence of health problems and unmet need for services differed by sub-populations.

Disability status was constructed based on responses about permanent functional limitations in the following domains: moving or walking, using arms or legs; seeing, even with glasses; speaking or communicating, even using sign language or other means; hearing, even with hearing aids; understanding or learning (concentrating and remembering); interacting with others, due to thoughts, feelings, emotions, or behaviors. For each domain, responses were yes or no. People were classified as having a disability if they answered affirmatively to at least one of the six domains (i.e., vision, hearing, mobility, cognition, interaction with others, and communication).

Indigenous status was classified based on mother-tongue language, a standard practice for analysis of ENAHO data [23]. People were considered to be indigenous if their mother-tongue was Quechua, Aymara or another native language, and non-indigenous if their mother-tongue was Spanish, or a foreign language. Since ethnicity is an inherent and stable characteristic that does not change over time, mother tongue can serve as a reliably proxy for identifying an individual's ethnicity [24].

Other stratification variables included: age group (0–17, 18–64, 65 or older), gender (male, female), health insurance coverage (yes, no), area of residence (urban, rural), and natural region (Coast, Highlands, and Jungle). Peru is divided into three natural regions: the Coast, which is located near the Pacific Ocean and includes the capital Lima and several other major cities; the Highlands, which includes the Andes mountainous region; and the Jungle, which comprises the more rural and geographically largest part of Peru, including the Peruvian Amazon.

## Statistical analysis

In the descriptive analysis, the characteristics of the included population were reported using absolute frequencies and weighted proportions. Additionally, comparisons of the two outcome variables before, during, and after the strict lockdown were performed. The period before the lockdown consisted of the four quarters of the years 2018 and 2019, and the first quarter of 2020; the period during the lockdown comprised from the second quarter of 2020 to the first quarter of 2021 (including national and targeted lockdowns); and the period after the lockdown was from the second quarter of 2021 to the second quarter of 2022 [25]. In addition, the differences in outcome variables during vs. before and after vs. before were evaluated using the Stata command *lincom* to calculate the t-statistic, p-value, and 95% confidence intervals (CI) of linear combinations of coefficients after previously conducted estimations.

Furthermore, an interrupted time series analysis was performed using the Stata command *itsa* to compare the two outcome variables before, during, and after the strict lockdown. We estimated the following ordinary least squares regression:

$$Y_t = \beta_0 + \beta_1 T_t + \beta_2 X_{1t} + \beta_3 X_{1t} T_{1t} + \beta_4 X_{2t} + \beta_5 X_{2t} T_{2t} + \in_t \tag{1}$$

In equation 1, $Y_t$ represents one of the two outcome variables measured in each quarter of the study period, and $T_t$ represents the time elapsed in each quarter since the beginning of the period. In this study, there were two "interventions": $X_{1t}$ represents the start of strict lockdown (second quarter of 2020), and $X_{2t}$ represents the end of strict lockdown (second quarter of 2021). $\beta_0$ represents the initial level of the outcome variable, $\beta_1$ represents the slope of the outcome variable before the start of lockdown, $\beta_2$ represents the immediate change that occurs after the implementation of lockdown,

$\beta_3$ represents the difference in the slopes of the outcome variable before and after the start of lockdown, $\beta_4$ represents the change in the outcome variable that occurs immediately after the end of lockdown, and $\beta_5$ represents the difference between the slopes of the variable for the first and second intervention. The analysis was run for the full population, and then for each of the included sub-groups separately.

Newey-West standard errors were applied to correct for autocorrelation [26]. To determine the appropriate number of lags for a model that accurately reflects the correct autocorrelation structure, the *actest* post-estimation command was used [26]. In all interrupted time series analyses, a lag of order one was used, as the p-value was >0.05, suggesting that the null hypothesis of no autocorrelation among the data cannot be rejected.

All analyses were conducted using the statistical software Stata v18.0 (Stata Corporation, College Station, Texas, USA). In addition, the complex design and sample weights of the ENAHO were taken into account using the *svy* command.

### Ethical considerations

All data are freely and publicly available [21]. The analysis received ethical approval from the London School of Hygiene & Tropical Medicine and the Universidad Peruana Cayetano Heredia.

### Results

The analysis included a total of 483,093 participants across survey waves. Characteristics of participants are shown in Table 1.

Pre-COVID, 57.1% of the population experienced a health problem in the preceding 4 weeks (Table 2). The likelihood of experiencing a health problem before COVID-19 increased with age, and was higher amongst women, indigenous people, people with disabilities, people living in rural areas or the Highlands region, and amongst those with insurance. During lockdowns, the proportion of the population experiencing a health problem decreased to 42.1%, a step change of -18.4 percentage points (pp) (Table 3 and Fig 1A). All groups experienced significant decreases in the likelihood of experiencing a health problem compared to pre-COVID. The average decrease was smaller for older adults, indigenous people, people with disabilities and people living in rural or Jungle areas, although there was no significant step change between groups (S1 Fig). After the removal of lockdowns, the proportion of the population experiencing a health problem increased slightly to 46.8%, a step change of -15.0pp compared to pre-pandemic levels. This level was similar for all groups. Compared to pre-pandemic, trend changes during lockdown and post-lockdown were overall and for almost all sub-groups positive but non-significant, indicating no clear pattern in trends over quarters of each period.

Amongst those with a health problem, 47.2% had an unmet need for healthcare pre-COVID-19 (Table 4). Unmet health needs were more common amongst working age adults, people with disabilities, indigenous people, people living in the Jungle and Highlands regions or rural areas and people with no health insurance. During lockdown, unmet health needs amongst people with a health problem increased to 61.0%, a step change of 16.7 pp (Table 5 and Fig 1B). The increase in unmet health needs was similar for all groups. Trend changes during lockdown were mostly negative but non-significant (with the exception of indigenous populations, people with health insurance, people with disabilities and people in Jungle areas), suggesting unmet health needs remained largely static across quarters during the lockdown period (S2 Fig). Unmet needs for healthcare then decreased slightly to 54.9% upon removal of lockdowns, with similar decreases for all groups. However, unmet need for healthcare still remained higher than pre-pandemic levels, with a step change of 13.8pp when comparing post-lockdown to pre-pandemic levels. Trend change for the post-lockdown period overall -1.6pp and was negative for all groups, suggesting a gradual decrease in unmet health needs over each quarter.

### Discussion

Overall, this analysis illustrates the significant impact of the COVID-19 pandemic on health and access to healthcare amongst the Peruvian population. Pre-COVID-19, more than half the population reported experiencing a health problem in

**Table 1. Characteristics of the included population, across 18 ENAHO waves (n = 483,093).**

| Characteristic | n | %* |
|---|---|---|
| Age group | | |
| 0-17 | 142,513 | 29·3 |
| 18-64 | 283,590 | 62·9 |
| 65 or older | 56,990 | 7·8 |
| Gender | | |
| Male | 235,432 | 50·1 |
| Female | 247,661 | 49·9 |
| Indigenous | | |
| No | 386,379 | 84·5 |
| Yes | 96,714 | 15·5 |
| Health insurance coverage | | |
| Yes | 388,977 | 77·3 |
| No | 94,116 | 22.7 |
| Disability | | |
| Yes | 24,859 | 4.1 |
| No | 458,234 | 95.9 |
| Health problem in the last 4 weeks | | |
| Yes | 259,501 | 50.8 |
| No | 223,592 | 49.2 |
| Area of residence | | |
| Urban | 304,166 | 78.9 |
| Rural | 178,927 | 21.1 |
| Natural region | | |
| Coast | 203,838 | 56.1 |
| Highlands | 171,592 | 31.0 |
| Jungle | 107,663 | 12.9 |

*The sample weights and the complex design of the ENAHO were included.

the previous 4 weeks, of whom almost half reported not receiving needed care. There were disparities amongst the population, with people with disabilities, indigenous people, people living in the Jungle, Highlands or rural areas experiencing a heightened risk of having a health problem and greater barriers to seeking care. During lockdowns, there was on average, a decrease of 18.4 pp in the prevalence of reported health problems. Groups at higher risk pre-COVID-19 experienced smaller declines. However, unmet needs for care increased 16.7 pp during this time. Post-lockdown, while there was a slight increase in people reporting a health problem compared to the lockdown period, levels remained lower than those observed before the pandemic. Similarly, unmet needs for healthcare amongst the sick decreased compared to the lockdown period but were still higher compared to pre-COVID-19.

Our results showed that the prevalence of health problems decreased by 18pp with the implementation of the lockdown. This finding is similar to previous studies which have shown that the impact of COVID-19 has led to a reduction in the incidence and prevalence of chronic diseases, infectious diseases, and accidents. In European countries such as Belgium [27], Finland [28] and the United Kingdom [29], the reported incidence of chronic diseases decreased during the first wave of the pandemic as a result of the lack of routine care, which made it difficult to diagnose these diseases. Although the reported incidence of these diseases fell dramatically, the lack of medical care led to increased mortality [29]. Similarly, infectious diseases showed a similar pattern in countries such as Japan [30], Germany [31], Netherlands [32],

**Table 2. Proportion of the population reporting a health problem in last 4 weeks before, during and after strict lockdown (n = 259,501).**

| Characteristic | Before lockdown | During lockdown | After lockdown | During vs. Before | After vs. Before |
|---|---|---|---|---|---|
| | % (95% CI) | % (95% CI) | % (95% CI) | pp (95% CI) | pp (95% CI) |
| Overall | 57.1 (56.6-57.6) | 42.1 (41.4-42.8) | 46.8 (46.1-47.6) | -15.0 (-15.9 to -14.1) *** | -10.3 (-11.2 to -9.4) *** |
| Age group | | | | | |
| 0-17 | 49.5 (48.8-50.3) | 30.5 (29.6-31.4) | 40.0 (39.1-40.9) | -19.1 (-20.2 to -18.0) *** | -9.5 (-10.7 to -8.4) *** |
| 18-64 | 58.7 (58.2-59.3) | 44.8 (44.0-45.7) | 47.8 (47.0-48.7) | -13.9 (-14.9 to -12.9) *** | -10.9 (-11.9 to -9.9) *** |
| 65 or older | 72.9 (72.0-73.9) | 64.3 (62.7-65.9) | 63.5 (62.0-65.0) | -8.6 (-10.5 to -6.8) *** | -9.4 (-11.2 to -7.7) *** |
| Gender | | | | | |
| Female | 61.5 (60.9-62.1) | 45.5 (44.6-46.3) | 51.0 (50.1-51.8) | -16.0 (-17.0 to -15.0) *** | -10.5 (-11.5 to -9.5) *** |
| Male | 52.8 (52.2-53.4) | 38.8 (38.0-39.6) | 42.7 (41.9-43.6) | -14.0 (-15.0 to -13.0) *** | -10.0 (-11.0 to -9.0) *** |
| Indigenous | | | | | |
| No | 55.2 (54.7-55.8) | 39.3 (38.5-40.0) | 44.1 (43.4-44.9) | -16.0 (-16.9 to -15.1) *** | -11.1 (-12.0 to -10.2) *** |
| Yes | 67.5 (66.4-68.5) | 57.1 (55.6-58.6) | 61.7 (60.4-63.1) | -10.3 (-12.1 to -8.6) *** | -5.7 (-7.3 to -4.1) *** |
| Health insurance coverage | | | | | |
| No | 54.8 (54.0-55.7) | 40.0 (38.8-41.1) | 42.5 (41.1-43.8) | -14.9 (-16.2 to -13.5) *** | -12.3 (-13.9 to -10.8) *** |
| Yes | 57.8 (57.3-58.4) | 42.8 (42.0-43.6) | 47.8 (47.1-48.6) | -15.0 (-16.0 to -14.1) *** | -10.0 (-11.0 to -9.1) *** |
| Disability | | | | | |
| No | 56.4 (55.9-57.0) | 41.2 (40.5-41.9) | 46.0 (45.2-46.7) | -15.2 (-16.1 to -14.3) *** | -10.4 (-11.3 to -9.5) *** |
| Yes | 73.6 (72.3-74.8) | 63.7 (61.6-65.8) | 66.1 (64.4-67.9) | -9.9 (-12.2 to -7.5) *** | -7.4 (-9.6 to -5.3) *** |
| Area of residence | | | | | |
| Rural | 60.5 (59.6-61.4) | 48.8 (47.5-50.0) | 56.1 (54.9-57.3) | -11.8 (-13.2 to -10.3) *** | -4.4 (-5.8 to -3.1) *** |
| Urban | 56.2 (55.5-56.8) | 40.4 (39.5-41.2) | 44.5 (43.7-45.4) | -15.8 (-16.8 to -14.8) *** | -11.6 (-12.7 to -10.6) *** |
| Natural region | | | | | |
| Coast | 54.5 (53.7-55.2) | 37.7 (36.7-38.8) | 40.4 (39.4-41.4) | -16.7 (-18.0 to -15.5) *** | -14.1 (-15.3 to -12.8) *** |
| Highlands | 63.3 (62.5-64.2) | 49.6 (48.4-50.8) | 56.9 (55.8-57.9) | -13.7 (-15.1 to -12.4) *** | -6.5 (-7.8 to -5.2) *** |
| Jungle | 53.6 (52.6-54.6) | 43.3 (42.0-44.7) | 51.5 (50.3-52.7) | -10.2 (-11.8 to -8.6) *** | -2.1 (-3.6 to -0.6) ** |

CI: confidence Interval, pp: percentage points; *** < 0.001, ** < 0.01, * < 0.05 p-values; Includes ENAHO sampling weights.

and the United Kingdom [33], where a significant reduction in the number of non-COVID-19 infectious disease notifications was observed. These infections have a similar mode of transmission to SARS-CoV-2, suggesting that social distancing measures contributed to controlling the spread not only of COVID-19 but also of other infectious diseases [33]. In addition, during the implementation of lockdown in several countries around the world, a reduction in road-traffic accidents and trauma hospitalizations was observed due to limited movement of people [34]. In this context, the measures taken to contain the spread of the SARS-CoV-2 virus and the shifting prioritization of health services in the early stages of the pandemic were the main reasons for the reduction in the prevalence of health problems. Additionally, people may not have known they had an unmet health need due to the lack of access to proper diagnoses and screening, as these services were disrupted in Peru [35,36].

Unmet healthcare needs amongst Peruvians increased during lockdowns and did not fully recover to pre-pandemic levels once lockdowns were removed. Other studies in China [37], the United States [38], Japan [39], Spain [40], South Korea [41], and Taiwan [42] have reported decreased utilisation of healthcare services during the initial waves of the pandemic. Additionally, a systematic review that encompassed 81 studies conducted in 20 countries worldwide, reporting a 37%

**Table 3. Step and trend changes in the percentage of people reporting a health problem during and after the COVID-19 lockdown, compared to the pre-lockdown period (n = 259,501).**

| Characteristic | Step change (During vs. Before) | Trend change (During vs. Before) | Step change (After vs Before) | Trend change (After vs Before) |
|---|---|---|---|---|
| Overall | -18.4 (-27.9 to -9.0) ** | 0.8 (-1.8-3.4) | -15.0 (-15.9 to -14.1) *** | 0.7 (-0.9-2.4) |
| Age group | | | | |
| 0-17 | -23.5 (-35.4 to -11.5) ** | 0.9 (-2.7-4.5) | -19.1 (-20.2 to -18.0) *** | 1.9 (-3.1-4.2) |
| 18-64 | -17.0 (-25.5 to -8.6) ** | 0.8 (-1.3-2.9) | -13.9 (-14.9 to -12.9) *** | 0.3 (-1.4-1.9) |
| 65 or older | -9.1 (-18.5 to 0.4) | -0.4 (-3.5-2.7) | -8.6 (-10.5 to -6.8) *** | 0.3 (-1.4-1.9) |
| Gender | | | | |
| Female | -20.1 (-29.0 to -11.1) *** | 1.2 (-1.7-4.1) | -16.0 (-17.0 to -15.0) *** | 0.9 (-0.6-2.5) |
| Male | -16.8 (-26.8 to -6.7) ** | 0.3 (-2.0-2.7) | -14.0 (-15.0 to -13.0) *** | 0.6 (-1.3-2.4) |
| Indigenous | | | | |
| No | -18.7 (-28.8 to -8.5) ** | 0.5 (-2.2-3.1) | -16.0 (-16.9 to -15.1) *** | 0.9 (-0.9-2.7) |
| Yes | -18.4 (-24.8 to -12.0) *** | 2.3 (-0.1-4.7) | -10.3 (-12.1 to -8.6) *** | -0.2 (-1.7-1.2) |
| Health insurance coverage | | | | |
| No | -18.7 (-30.3 to -7.4) ** | 0.9 (-1.6-3.4) | -14.9 (-16.2 to -13.5) *** | 0.04 (-2.1-2.2) |
| Yes | -18.2 (-27.2 to -9.3) ** | 0.7 (-1.9-3.3) | -15.0 (-16.0 to -14.1) *** | 0.8 (-0.7-2.4) |
| Disability | | | | |
| No | -18.6 (-28.2 to -9.1) ** | 0.7 (-1.8-3.3) | -15.2 (-16.1 to -14.3) *** | 0.8 (-0.9-2.5) |
| Yes | -11.3 (-19.4 to -3.2) * | 0.2 (-3.9-4.3) | -9.9 (-12.2 to -7.5) *** | 0.05 (-1.7-1.8) |
| Area of residence | | | | |
| Rural | -20.6 (-28.0 to -13.2) *** | 2.3 (-0.4-5.1) | -11.8 (-13.2 to -10.3) *** | 0.9 (-0.4-2.2) |
| Urban | -17.8 (-27.9 to -7.7) ** | 0.4 (-2.3-3.0) | -15.8 (-16.8 to -14.8) *** | 0.7 (-1.1-2.5) |
| Natural region | | | | |
| Coast | -16.2 (-28.3 to -4.2) * | -0.7 (-3.6-2.2) | -16.7 (-18.0 to -15.5) *** | 1.0 (-1.2-3.2) |
| Highlands | -23.4 (-30.2 to -16.6) *** | 3.1 (0.3-5.9) * | -13.7 (-15.1 to -12.4) *** | -0.04 (-1.4-1.3) |
| Jungle | -15.8 (-21.9 to -9.8) *** | 1.7 (0.01-3.4) * | -10.2 (-11.8 to -8.6) *** | 1.7 (0.3-3.1) * |

95% confidence intervals are shown in parentheses; ***<0.001, **<0.01, *<0.05 p-values; Includes ENAHO sampling weights.

reduction in all healthcare services (visits, admissions, diagnosis, and treatment) during the pandemic compared to the pre-pandemic period [4]. Specifically, the systematic review indicated a 42% reduction in healthcare visits, a 28% reduction in access to diagnostic tests, and a 30% reduction in access to therapeutic measures. However, in these studies it is not known if decreased utilization is arising from decreased need for services or increased barriers to access. This study from Peru indicates a combination of both may be at play, with health needs decreasing but barriers to accessing required care amongst people with a health problem increasing. In Peru, there was a significant reallocation of services towards COVID-19 care, and mitigation measures (e.g., telemedicine) appear to have been inadequate to fill demand [35,36].

Certain groups have had higher health needs and unmet needs health services both pre- and during the pandemic. For example, people with disabilities have historically been more likely to have worse health and greater unmet care needs than people without disabilities, due for example to higher levels of poverty, poor accessibility of health facilities, discrimination and low availability of disability-related services [18]. Then COVID-19 may have worsened access, due for example to a de-prioritisation of disability-related healthcare. Similarly, indigenous groups and people living in rural locations also have historically had poorer access to healthcare and then experienced worse health outcomes during COVID-19, due to higher levels of poverty, far distances to health facilities, and for indigenous groups, lack of intercultural care [14,19]. The Peruvian health system has issues with fragmentation, which leads to economic resources being distributed inequitably

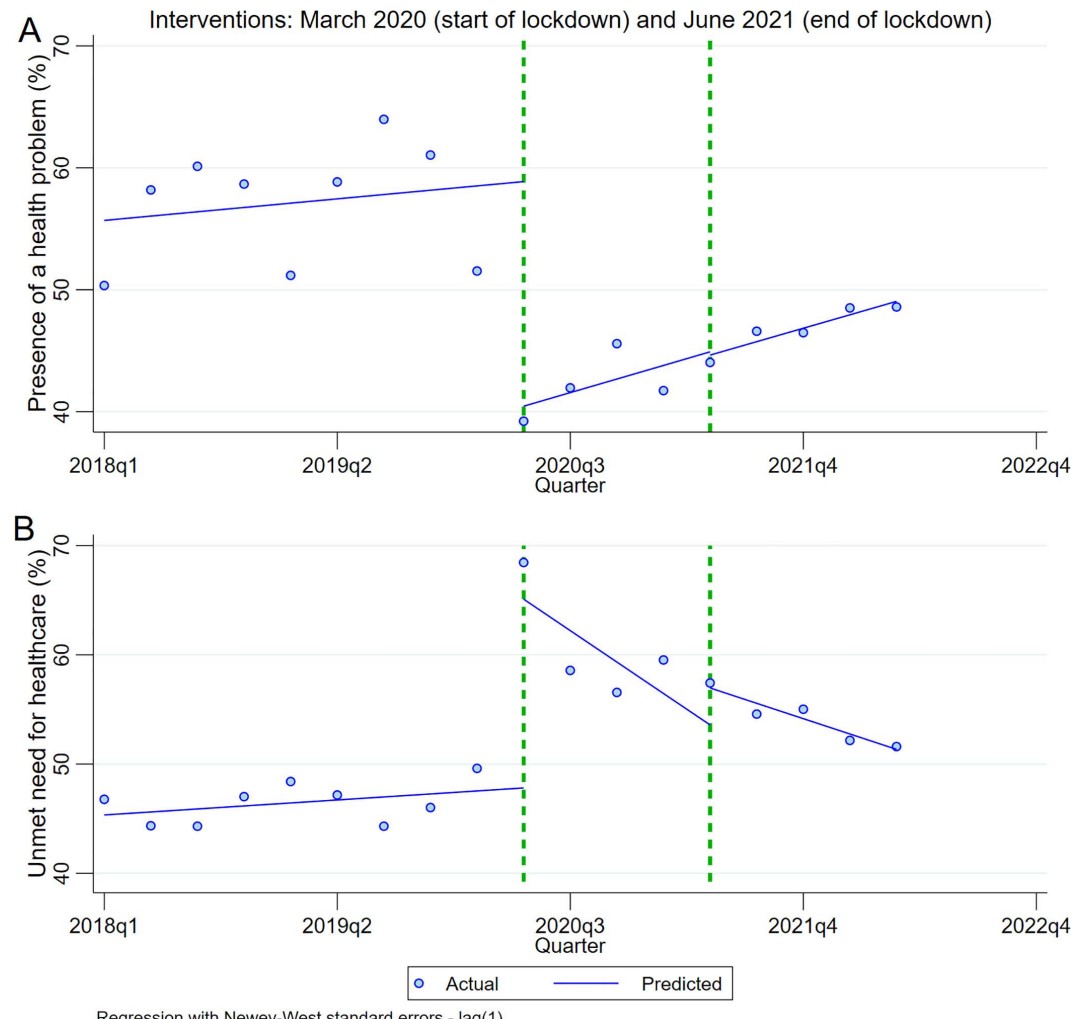

**Fig 1. Interrupted time series analysis of the impact of COVID-19 lockdown on (A) presence of a health problem in last 4 weeks and (B) unmet need for healthcare.**

across different regions [14]. Finally, people without health insurance also had a higher likelihood of unmet health needs during all three periods, likely due to financial barriers to healthcare access that may have been exacerbated by severe job losses triggered by COVID-19. For example, seven million Peruvians were newly out of work in 2020, increasing financial barriers to seeking care [13].

## Strengths and limitations

One of the main strengths of this study is the use of a nationally representative, quarterly database, which allows our results to be extrapolated to the entire Peruvian population and to conduct an interrupted time series analysis with multiple data points. It also explores both the likelihood of experiencing a health problem and unmet needs for healthcare amongst the sick – the use of both outcomes adds to the literature, as most other analyses have focused on changes in healthcare utilisation without consideration of the causes. Similarly, it explores both the effect of lockdowns as well as the removal of lockdowns, to investigate whether outcomes returned to pre-pandemic levels.

**Table 4. Comparison of the proportion of unmet needs for healthcare before, during and after strict lockdown for the COVID-19 pandemic (n = 259,501).**

| Characteristic | Before lockdown % (95% CI) | During lockdown % (95% CI) | After lockdown % (95% CI) | During vs. Before pp (95% CI) | After vs. Before pp (95% CI) |
|---|---|---|---|---|---|
| Overall | 47.2 (46.6-47.8) | 61.0 (60.0-62.0) | 54.9 (54.1-55.7) | 13.8 (12.6-14.9) *** | 7.7 (6.7-8.7) *** |
| Age group | | | | | |
| 0-17 | 45.7 (44.8-46.6) | 62.3 (60.6-63.9) | 54.6 (53.3-55.9) | 16.5 (14.7-18.3) *** | 8.9 (7.3-10.5) *** |
| 18-64 | 48.0 (47.3-48.6) | 60.4 (59.3-61.5) | 54.7 (53.8-55.6) | 12.4 (11.2-13.7) *** | 6.7 (5.6-7.8) *** |
| 65 or older | 46.2 (45.2-47.2) | 62.0 (60.3-63.7) | 56.7 (55.2-58.1) | 15.8 (13.8-17.8) *** | 10.5 (8.7-12.2) *** |
| Gender | | | | | |
| Female | 46.8 (46.2-47.5) | 60.8 (59.6-61.9) | 54.4 (53.5-55.3) | 13.9 (12.6-15.2) *** | 7.6 (6.5-8.7) *** |
| Male | 47.6 (46.9-48.4) | 61.3 (60.1-62.5) | 55.4 (54.4-56.4) | 13.6 (12.3-15.0) *** | 7.8 (6.6-9.0) *** |
| Indigenous | | | | | |
| No | 45.5 (44.9-46.1) | 59.0 (57.9-60.1) | 53.3 (52.5-54.2) | 13.5 (12.2-14.7) *** | 7.8 (6.8-8.9) *** |
| Yes | 55.0 (53.8-56.1) | 68.3 (66.7-69.8) | 60.9 (59.6-62.3) | 13.3 (11.4-15.2) *** | 6.0 (4.2-7.7) *** |
| Health insurance coverage | | | | | |
| No | 52.4 (51.4-53.3) | 63.9 (62.2-65.6) | 58.6 (56.9-60.2) | 11.5 (9.6-13.5) *** | 6.2 (4.3-8.1) *** |
| Yes | 45.6 (45.0-46.3) | 60.1 (59.1-61.1) | 54.2 (53.3-55.0) | 14.5 (13.3-15.7) *** | 8.5 (7.5-10.0) *** |
| Disability | | | | | |
| No | 47.1 (46.5-47.7) | 60.9 (59.8-61.9) | 54.7 (53.9-55.5) | 13.8 (12.6-15.0) *** | 7.6 (6.6-8.6) *** |
| Yes | 49.4 (47.9-50.9) | 63.0 (60.7-65.2) | 57.6 (55.5-59.8) | 13.6 (10.9-16.2) *** | 8.2 (5.6-10.8) *** |
| Area of residence | | | | | |
| Rural | 56.4 (55.5-57.4) | 69.9 (68.5-71.2) | 64.2 (63.0-65.4) | 13.4 (11.9-15.0) *** | 7.8 (6.3-9.2) *** |
| Urban | 44.4 (43.7-45.1) | 58.2 (57.0-59.4) | 51.9 (51.0-52.9) | 13.8 (12.4-15.1) *** | 7.5 (6.3-8.7) *** |
| Natural region | | | | | |
| Coast | 42.2 (41.4-43.0) | 54.8 (53.2-56.4) | 49.1 (47.9-50.4) | 12.6 (10.9-14.4) *** | 7.0 (5.5-8.5) *** |
| Highlands | 53.2 (52.2-54.1) | 68.5 (67.2-69.8) | 60.9 (59.7-62.0) | 15.3 (13.8-16.9) *** | 7.7 (6.2-9.1) *** |
| Jungle | 52.3 (51.2-53.4) | 63.9 (62.2-65.7) | 59.0 (57.7-60.4) | 11.6 (9.6-13.7) *** | 6.7 (5.0-8.4) *** |

CI: confidence Interval, pp: percentage points; ***<0.001, **<0.01, *<0.05 p-values; Includes ENAHO sampling weights.

However, this study has some limitations. For example, unmet need for healthcare is not assessed objectively. It is possible that some individuals perceived that they required health services for their health problems when it was in fact not required (e.g., illness would resolve without medical intervention), which could lead to overestimation of unmet need. Alternatively, and potentially more likely, unmet healthcare needs may have been underestimated as people either did not think they required healthcare when they did or received inappropriate healthcare. For example, some individuals may not have recognized their need for healthcare due to low health literacy. Even people who sought healthcare may not have received care that was appropriate for resolving their health condition. For example, they may have sought pain relief at a pharmacy instead of investigating or treating the underlying cause at a health center, or they may have visited a health facility where the required services or medications were not available or of sub-optimal quality. Further, the wording of the question on health needs focuses on more urgent or new health needs (e.g., due to illness, accident, relapse of a chronic condition). The available data may not have adequately captured needs for more routine and ongoing healthcare (e.g., management of a pre-existing chronic condition or disability) or for preventative care (e.g., vaccines, primary care, cancer screenings). These biases are likely non-differential across the three time periods, although they may be differential amongst sub-groups. For instance, people with lower education levels and limited previous interactions with health

**Table 5. Step and trend changes in the percentage of unmet needs for healthcare during and after the COVID-19 lockdown, compared to the pre-lockdown period (n = 259,501).**

| Characteristic | Step change (During vs. Before) | Trend change (During vs. Before) | Step change (After vs Before) | Trend change (After vs Before) |
|---|---|---|---|---|
| Overall | 16.7 (11.0-22.5) *** | -3.0 (-6.3-0.2) | 13.8 (12.6-14.9) *** | -1.6 (-2.3 to -1.0) *** |
| Age group | | | | |
| 0-17 | 18.3 (12.4-24.2) *** | -2.7 (-6.0-0.6) | 16.5 (14.7-18.3) *** | -2.7 (-3.9 to -1.6) *** |
| 18-64 | 15.5 (9.8-21.3) *** | -3.0 (-6.2-0.2) | 12.4 (11.2-13.7) *** | -1.2 (-1.9 to -0.4) ** |
| 65 or older | 20.2 (13.1-27.4) *** | -3.5 (-7.1-0.1) | 15.8 (13.8-17.8) *** | -1.5 (-2.4 to -0.7) ** |
| Gender | | | | |
| Female | 17.3 (11.3-23.3) *** | -3.2 (-6.7-0.2) | 13.9 (12.6-15.2) *** | -1.8 (-2.5 to -1.2) *** |
| Male | 16.1 (10.6-21.6) *** | -2.8 (-5.8-0.3) | 13.6 (12.3-15.0) *** | -1.4 (-2.0 to -0.7) ** |
| Language spoken | | | | |
| Non-native | 16.4 (10.2-22.6) *** | -2.9 (-6.4-0.7) | 13.5 (12.2-14.7) *** | -1.5 (-2.1 to -0.9) *** |
| Native | 16.6 (12.5-20.7) *** | -3.7 (-5.9 to -1.6) ** | 13.3 (11.4-15.2) *** | -1.9 (-2.8 to -0.9) ** |
| Health insurance coverage | | | | |
| No | 12.3 (5.6-19.0) *** | -2.2 (-5.8-1.3) | 11.5 (9.6-13.5) *** | -0.5 (-1.6-0.6) |
| Yes | 17.9 (12.4-23.5) *** | -3.2 (-6.4 to -0.8) * | 14.5 (13.3-15.7) *** | -1.8 (-2.4 to -1.2) *** |
| Type of health insurance | | | | |
| No insurance | 12.3 (5.6-19.0) *** | -2.2 (-5.8-1.3) | 11.5 (9.6-13.5) *** | -0.5 (-1.6-0.6) |
| Armed forces and Police | 18.1 (13.3-22.8) *** | -3.6 (-5.6 to -1.6) ** | 16.5 (9.3-23.7) *** | -1.0 (-4.6-2.5) |
| Public Health insurance | 16.7 (10.7-22.7) *** | -3.2 (-6.7-0.4) | 13.0 (11.7-14.3) *** | -2.0 (-2.5 to -1.4) *** |
| Social Health insurance | 19.5 (14.6-24.3) *** | -3.3 (-5.9 to -0.8) * | 15.8 (13.9-17.8) *** | -1.3 (-2.0 to -0.6) ** |
| Private insurance | 39.5 (26.5-52.5) *** | -14.7 (-18.2 to -11.2) *** | 20.7 (10.6-30.9) *** | -2.1 (-6.9-2.6) |
| Disability | | | | |
| No | 16.7 (10.9-22.5) *** | -2.9 (-6.2-0.4) | 13.8 (12.6-15.0) *** | -1.6 (-2.2 to -0.9) *** |
| Yes | 16.5 (12.0-21.0) *** | -4.7 (-7.0 to -2.3) ** | 13.6 (10.9-16.2) *** | -2.7 (-4.3 to -1.2) ** |
| Area of residence | | | | |
| Rural | 15.6 (10.0-21.3) *** | -2.8 (-5.7-0.6) | 13.4 (11.9-15.0) *** | -2.3 (-3.0 to -1.6) *** |
| Urban | 17.1 (11.3-23.0) *** | -3.2 (-6.6-0.2) | 13.8 (12.4-15.1) *** | -1.4 (-2.0 to -0.8) *** |
| Natural region | 17.1 (11.3-22.9) *** | -3.4 (-7.0-0.2) | 12.6 (10.9-14.4) *** | -3.3 (-6.7-0.7) |
| Coast | | | | |
| Highlands | 16.7 (11.0-22.5) *** | -2.4 (-5.2-0.4) | 15.3 (13.8-16.9) *** | -2.2 (-3.0 to -1.3) *** |
| Jungle | 16.6 (9.5-23.7) *** | -5.2 (-8.5 to -1.9) ** | 11.6 (9.6-13.7) *** | -3.1 (-4.2 to -2.1) *** |

95% confidence intervals are shown in parentheses; ***<0.001, **<0.01, *<0.05 p-values; Includes ENAHO sampling weights.

systems often have lower health literacy, affecting recognition of health problems and where to seek appropriate care. Meanwhile people in rural and remote areas or who live in poverty, and groups such as people with disabilities, Indigenous people and people who are uninsured, face additional geographic, financial and attitudinal barriers to receiving quality healthcare [19,43–45]. As such, some inequalities between sub-groups may be underestimated.

Another limitation is that the transition from face-to-face interviews to telephone interviews during the COVID-19 lockdown period may have introduced reporting bias. Although response rates were high for all reporting periods – including periods with telephone interviews – the quality of data collection may have been more limited with phone interviews compared to face-to-face. These limitations may reduce comparability across time periods and could contribute to measurement variability in the estimated prevalence of unmet need for healthcare.

## Conclusion

This research carries important implications for preparedness for future pandemics and other public health crises, as well as for the routine operation of health systems in Peru and elsewhere. It is clear that both before and during COVID-19, certain groups (e.g., people with disabilities, Indigenous people and people in rural and remote areas) experienced worse health outcomes and poorer access to care. As such, more targeted approaches are needed to address the persistent barriers faced by these groups, such as through improving the geographic availability, affordability and accessibility of health services. Furthermore, the needs of specific groups at risk of exclusion must be considered in pandemic preparedness planning in Peru and other contexts.

## Supporting information

**S1 Fig. Interrupted time series analysis of the impact of COVID-19 lockdown on the presence of a health problem in last 4 weeks, stratified by sociodemographic variables.**
(PDF)

**S2 Fig. Interrupted time series analysis of the impact of COVID-19 lockdown on unmet need for healthcare, stratified by sociodemographic variables.**
(PDF)

## Author contributions

**Conceptualization:** Shanquan Chen, Francisco Diez-Canseco, Pamela Smith, Xanthe Hunt, Lena Morgon Banks.

**Data curation:** Rodrigo Vargas-Fernández, Akram Hernández-Vásquez.

**Formal analysis:** Rodrigo Vargas-Fernández, Akram Hernández-Vásquez.

**Funding acquisition:** Lena Morgon Banks.

**Writing – original draft:** Rodrigo Vargas-Fernández, Akram Hernández-Vásquez, Lena Morgon Banks.

**Writing – review & editing:** Rodrigo Vargas-Fernández, Akram Hernández-Vásquez, Shanquan Chen, Francisco Diez-Canseco, Pamela Smith, Xanthe Hunt, Lena Morgon Banks.

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
