## [Decision Letter · Decision Letter 0]

11 Oct 2024

PGPH-D-24-01626

Impact of COVID-19 on unmet needs for healthcare in Peru: an interrupted time series analysis

Dear Dr. Vargas-Fernández,

Thank you for submitting your manuscript to PLOS Global Public Health. After careful consideration, we feel that it has merit but does not fully meet PLOS Global Public Health’s publication criteria as it currently stands. Therefore, we invite you to submit a revised version of the manuscript that addresses the points raised during the review process.

The manuscript has been evaluated by two reviewers, and their comments are available below.

Reviewer 1 is enthusiastic about your paper and the value of your research, but the reviewer notes a lack of familiarity with interrupted time series analyses. Reviewer 2 raises some concerns regarding the details of these analyses and requests clarification.

Could you please carefully revise the manuscript to address all comments raised?

We look forward to receiving your revised manuscript.

Kind regards,

Steve Zimmerman, PhD

PLOS Staff Editor

Journal Requirements:

1. Your current Financial Disclosure states, “PENDA project PO8073)”. However, it is missing in your funding information on the submission form. Please indicate by return email the full and correct funding information for your study and confirm the order in which funding contributions should appear. Please be sure to indicate whether the funders played any role in the study design, data collection and analysis, decision to publish, or preparation of the manuscript.

Additional Editor Comments (if provided):

Reviewers' comments:

Reviewer's Responses to Questions

**Comments to the Author**

1. Does this manuscript meet PLOS Global Public Health’s publication criteria? Is the manuscript technically sound, and do the data support the conclusions? The manuscript must describe methodologically and ethically rigorous research with conclusions that are appropriately drawn based on the data presented.

Reviewer #1: No

Reviewer #2: Yes

2. Has the statistical analysis been performed appropriately and rigorously?

Reviewer #1: No

Reviewer #2: Yes

3. Have the authors made all data underlying the findings in their manuscript fully available (please refer to the Data Availability Statement at the start of the manuscript PDF file)?

Reviewer #1: Yes

Reviewer #2: Yes

4. Is the manuscript presented in an intelligible fashion and written in standard English?

Reviewer #1: Yes

Reviewer #2: Yes

5. Review Comments to the Author

Reviewer #1: Methods:

- Does the parametrization includes a vector of confounding covariates?

- The current parametrization lacks of addressing serial correlation in the error terms. Newey West variance estimator is advised.

Results:

- Figures are not available for review. It would be great to display a visualization of the empirical distribution of the outcomes, the fitted and predicted regression lines.

- The coefficients table look a bit odd. The binary variables report two coefficients without setting a reference level.

Discussion:

- This section should be enhanced by connecting the findings with the real impact in the Peruvian Healthcare system. For now, that connection is lacking.

Reviewer #2: This paper provides an exceptional analysis of the impact of COVID-19 safety measures in Peru. It presents valuable data from a low-middle income country (LMIC), a rarity in current literature. The primary aim of this study was to assess the impact of COVID-19 on unmet healthcare needs across Peru. This analysis will be highly useful for preparing for future pandemics in similar contexts.

The study focuses on understanding unmet health needs before and after lockdowns. An interrupted time series analysis was conducted using the National Household Survey on Living Conditions and Poverty (ENAHO) data from 2018 to 2022. ENAHO is a population-based survey managed by the National Institute of Statistics and Informatics of Peru.

The use of data collected through both face-to-face and telephone interviews with trained interviewers is a major strength of this study. The total and annual sample sizes are strong, and the response rate is impressive, adding reliability to the findings.

The dependent variables are well-defined, and the analysis is stratified based on disability status and indigenous status. Other variables, such as age, gender, health insurance coverage, and geographical regions, are also included and thoroughly described, showcasing the comprehensiveness of the data.

Statistical analyses are meticulously detailed and disaggregated. Although I am not an expert in the mathematical modeling of interrupted time series, the methodology appears sound and consistent with other high-quality research.

The results are presented clearly, and the discussion is compelling. The use of graphical presentations for the interrupted time series could further enhance the visual understanding of the findings.

The limitations are transparently presented and appropriately addressed.

Overall, I find this manuscript exceptionally informative and well-written. It merits publication.

6. PLOS authors have the option to publish the peer review history of their article (what does this mean?). If published, this will include your full peer review and any attached files.

**Do you want your identity to be public for this peer review?** For information about this choice, including consent withdrawal, please see our Privacy Policy.

Reviewer #1: No

Reviewer #2: No

---

## [Editor Report · Decision Letter 1]

11 Aug 2025

PGPH-D-24-01626R3 

Impact of COVID-19 on unmet needs for healthcare in Peru: an interrupted time series analysis 

Dear Dr. Vargas-Fernández:

I'm pleased to inform you that your manuscript has been deemed suitable for publication in PLOS Global Public Health. Congratulations! Your manuscript is now with our production department. 

If your institution or institutions have a press office, please let them know about your upcoming paper now to help maximize its impact. If they'll be preparing press materials, please inform our press team within the next 48 hours. Your manuscript will remain under strict press embargo until 2 pm Eastern Time on the date of publication. For more information please contact globalpubhealth@plos.org.

If we can help with anything else, please email us at globalpubhealth@plos.org. 

Thank you for submitting your work to PLOS Global Public Health and supporting open access. 

Kind regards, 

PLOS Global Public Health Editorial Office Staff

on behalf of

Dr. Ejemai Eboreime 

Academic Editor

---

## [Decision Letter · Decision Letter 2]

18 Jun 2025

PGPH-D-24-01626R2

Impact of COVID-19 on unmet needs for healthcare in Peru: an interrupted time series analysis

Dear Dr. Vargas-Fernández,

Thank you for submitting your manuscript to PLOS Global Public Health. After careful consideration, we feel that it has merit but does not fully meet PLOS Global Public Health’s publication criteria as it currently stands. Therefore, we invite you to submit a revised version of the manuscript that addresses the points raised during the review process.

We look forward to receiving your revised manuscript.

Kind regards,

Ejemai Eboreime, MD, MSc, PhD

Academic Editor

Journal Requirements:

Additional Editor Comments (if provided):

Clarify measurement of unmet need: The operationalization of "unmet healthcare need" could be interpreted variably. A clearer explanation and possible limitations (e.g., care sought in non-biomedical facilities) should be emphasized in the limitations section.

Limitations Expansion: While some limitations are acknowledged, additional discussion around the effect of changes in data collection mode (e.g., shift to phone interviews) and potential reporting bias during lockdown periods would be valuable.

Reviewers' comments:

Reviewer's Responses to Questions

**Comments to the Author**

1. If the authors have adequately addressed your comments raised in a previous round of review and you feel that this manuscript is now acceptable for publication, you may indicate that here to bypass the “Comments to the Author” section, enter your conflict of interest statement in the “Confidential to Editor” section, and submit your "Accept" recommendation.

Reviewer #3: (No Response)

2. Does this manuscript meet PLOS Global Public Health’s publication criteria? Is the manuscript technically sound, and do the data support the conclusions? The manuscript must describe methodologically and ethically rigorous research with conclusions that are appropriately drawn based on the data presented.

Reviewer #3: Partly

3. Has the statistical analysis been performed appropriately and rigorously?

Reviewer #3: Yes

4. Have the authors made all data underlying the findings in their manuscript fully available (please refer to the Data Availability Statement at the start of the manuscript PDF file)?

Reviewer #3: Yes

5. Is the manuscript presented in an intelligible fashion and written in standard English?

Reviewer #3: Yes

6. Review Comments to the Author

Reviewer #3: Write the abstract in the correct order (background, objectives, methods, results, and conclusion).

Make table 3 more understandable or merge with table 2 if possible.

The discussion is good, but the conclusion looks like a recommendation.

7. PLOS authors have the option to publish the peer review history of their article (what does this mean?). If published, this will include your full peer review and any attached files.

**Do you want your identity to be public for this peer review?** For information about this choice, including consent withdrawal, please see our Privacy Policy.

Reviewer #3: No

---

## [Editor Report · Decision Letter 3]

23 Jul 2025

Impact of COVID-19 on unmet needs for healthcare in Peru: an interrupted time series analysis

PGPH-D-24-01626R3

Dear Dr. Vargas-Fernández,

We are pleased to inform you that your manuscript 'Impact of COVID-19 on unmet needs for healthcare in Peru: an interrupted time series analysis' has been provisionally accepted for publication in PLOS Global Public Health.

Best regards,

Ejemai Eboreime, MD, MSc, PhD

Academic Editor